# REX: RAPID EXPLORATION AND EXPLOITATION FOR AI AGENTS

## ABSTRACT

AI agents leveraging the capabilities of Large Language Models (LLMs) and Reinforcement Learning (RL) techniques have garnered growing attention due to their commendable performance in autonomously executing real-world tasks. Effective exploration of the action space is paramount for the successful accomplishment of diverse tasks by these AI agents. In this paper, we propose an enhanced approach for **R**apid **E**xploration and e**X**ploitation of action space for LLM-based AI agents, called **REX**. Existing LLM-driven agents have inherent limitations, such as a heavy reliance on precise descriptions for decision-making, and the lack of a systematic approach to leverage try-and-fail procedures akin to traditional RL. REX introduces an additional layer of rewards and integrates concepts similar to Upper Confidence Bound (UCB) scores, leading to more robust and efficient AI agent performance. This approach has the advantage of enabling the utilization of offline behaviors from logs and allowing seamless integration with existing foundation models while it does not require any model fine-tuning. Through comparative analysis with existing methods such as Chain-of-Thought (CoT) and Reflexion, REX-based methods demonstrate comparable performance and, in certain cases, even surpass the results achieved by these existing techniques. Notably, REX-based methods exhibit remarkable reductions in execution time while systematically exploring the action space of AI agents, enhancing their practical applicability across a diverse set of scenarios.

## 1 INTRODUCTION

AI agents driven by Large Language Models (LLMs) have become a very active research topic recently. A series of applications, such as AutoGPT (Gravitas, 2023), BabyAGI (Nakajima, 2023), AgentGPT (age, 2023), WebGPT (Nakano et al., 2022), OpenAGI (Ge et al., 2023), MetaGPT Hong et al. (2023), etc. have been proposed and well adopted in various application scenarios. AI agents built upon LLMs typically transform user inputs into a standardized prompt using predefined templates. These templates generally necessitate users to furnish comprehensive descriptions of their task objectives and plans. This framework allows the language model to comprehend the action space effectively. In the context of LLM-based AI agents, one crucial aspect of effective engineering lies in explicitly specifying the prompt. This specification serves as a blueprint for shaping the LLM's output format in a structured manner. This structured format enables the system to precisely extract the actions chosen by the model. Subsequently, based on the agent's actions, the system operates similarly to conventional reinforcement learning setup by executing state transitions and offering potential rewards or feedback to the agent.

Despite the significant accomplishments of LLM-based AI agents across various application domains, it is evident that they are still in a nascent stage of development (Kaddour et al., 2023; Zhang et al., 2023), leaving ample room for enhancement. One specific area that warrants meticulous attention and refinement pertains to the incorporation of feedback and rewards to augment their decision-making prowess. Prominent RL algorithms, such as Policy Gradient (Sutton et al., 2000), Proximal Policy Optimization Algorithm (PPO) (Schulman et al., 2017), Trust Region Policy Optimization (TRPO) (Schulman et al., 2015), and Advantage Actor Critic methods (Mnih et al., 2016) have the capability to update the model parameters of these agents based on the feedback and rewards emanating from their environments. However, it is important to note that the process of updating the model parameters in the context of LLM-based AI agents entails substantial data

collection, is time-intensive, and incurs significant costs. Furthermore, given that each unique environment may necessitate distinct adjustments, fine-tuning LLMs for every environment could be challenging especially for real-world applications. Even minor alterations in the task setup, the environment, or the action space can mandate a complete re-fine-tuning of the entire LLM. In light of recent empirical discoveries underscoring the effectiveness of In-Context Learning (ICL) (Dong et al., 2023; Dai et al., 2023; Akyürek et al., 2023; Pan et al., 2023), numerous solutions based on ICL have been put forth. Nevertheless, ICL-based LLM agents confront several limitations: (1) **Lack of systematic rewards incorporation:** While they can generate actions based on input data, they often struggle to systematically integrate reward signals, hindering their ability to learn and optimize their performance; (2) **Exploration-Exploitation trade-off:** LLMs face formidable challenges in striking the delicate balance between exploration and exploitation. Achieving optimal performance necessitates exploring novel strategies to unearth potentially superior rewards while simultaneously exploiting existing knowledge to maximize immediate gains; (3) **Accuracy & Latency:** In complex problem-solving scenarios with multiple sequential steps, requiring LLMs to generate all the steps simultaneously proves problematic. This approach not only results in inaccuracies due to the expansive action space and myriad states involved but also necessitates more extensive instructions within the prompt. Furthermore, invoking the LLM separately for each intermediate step in the solution would incur significant time overheads, rendering it a time-consuming endeavor.

To overcome these constraints, this paper introduces an novel approach, **R**apid **E**xploration and e**X**ploitation for AI agents **(REX)**, that is designed to empower LLMs with the capability to seamlessly integrate rewards into their models and effectively manage the exploration-exploitation trade-off, all while significantly accelerating the overall learning process. In summary, our contributions can be encapsulated as follows: (1) **Integration of rewards into the prompt utilizing the Upper Confidence Bound (UCB)** framework, thereby enabling a systematic exploration and exploitation of the action space; (2) **Concurrent pursuit of exploration and exploitation across all steps of the solution**, ensuring swift progress without compromising accuracy; (3) **Systematically influencing the logits of LLMs through what we term UCL (UCB applied for Logits of LLM)**, providing enhanced control over the generation of actions.

## 2 RELATED WORK

A significant amount of effort have been made to harness LLMs to build AI agents. CoT (Wei et al., 2022) encourages the model to solve the task step-by-step before arriving at the final answer. ReAct (Yao et al., 2023) advocates utilizing the reasoning and action capabilities of LLMs to promote interactive engagement with the environment, exemplified by the utilization of the Wikipedia search API. On the other hand, Reflexion (Shinn et al., 2023) collects feedback i.e. reflects on it's decision making and then improvises. The RAP system (Hao et al., 2023) transforms the LLM into a dual-purpose entity, serving as both a world model and a reasoning agent. It integrates Monte Carlo Tree Search for strategic exploration within the expansive domain of reasoning, guided by environmental rewards. These techniques has inspired various applications like AutoGPT (Yang et al., 2023), BabyAGI (Nakajima, 2023), WebGPT (Nakano et al., 2021), HuggingGPT (Shen et al., 2023), Generative agents (Park et al., 2023), Langchain (Chase, 2023), etc.

Nonetheless, the majority of these methodologies exhibit limitations in effectively assimilating environmental feedback and optimizing action selection from the available action space. Among these techniques, Reflexion and RAP partially address this issue. Reflexion(+CoT), endeavors to ameliorate this challenge by constructing a rationale for decision-making. However, when confronted with actions of considerable similarity, there exists a notable risk of model confusion. In contrast, RAP exhibits a commendable ability to judiciously select actions through the utilization of Monte Carlo Tree Search (MCTS). Nevertheless, its reliance on the retrieval of token logit values for reward calculation precludes the utilization of widely adopted APIs such as ChatGPT. Additionally, RAP necessitates a significantly higher frequency of queries to the LLM compared to alternative techniques. Conversely, REX offers a distinct advantage by employing UCB scores to unequivocally identify actions. This approach substantially reduces the likelihood of ambiguity among actions within the action space and accelerates the decision-making process, as described in section 4.

## 3   Monte Carlo Tree Search

In this section we will discuss Monte Carlo Tree Search (MCTS) which is the backbone of REX. In recent years, MCTS has gained significant attention as a powerful algorithm for decision-making in various domains, particularly in game-playing applications. MCTS has demonstrated remarkable performance in games such as Chess, Go, and Poker, surpassing human expertise in some cases. However, its applications are not limited to games alone, as MCTS can be adapted to tackle complex decision-making problems in diverse fields. At its core, MCTS is a simulation-based search algorithm that leverages random sampling to explore the search space and make informed decisions. It combines elements of tree search and Monte Carlo sampling to iteratively build a search tree and estimate the value of each possible action. The main steps of the MCTS algorithm can be summarized as follows: (1) **Selection:** Starting from the root of the search tree, the algorithm traverses down the tree by selecting actions that balance exploration and exploitation. It uses a selection policy, such as the Upper Confidence Bound, to determine the most promising nodes; (2) **Expansion:** Once a leaf node is reached, the algorithm expands the tree by randomly picking a node from a set of possible child nodes; (3) **Simulation:** MCTS performs Monte Carlo simulations (also known as rollouts) from newly expanded node. These simulations play out random sequences of actions until reaching a terminal state, yielding an outcome or reward; (4) **Backpropagation:** After a simulation is completed, the result is backpropagated up the tree. The statistics of the visited nodes, such as the number of visits and accumulated rewards, are updated accordingly. These four steps are repeated iteratively for a specified number of iterations or until a time limit is reached. As more iterations are performed, the search tree evolves and provides increasingly accurate estimates of the value of different actions. Figure 1 shows the pictorial representation of MCTS.

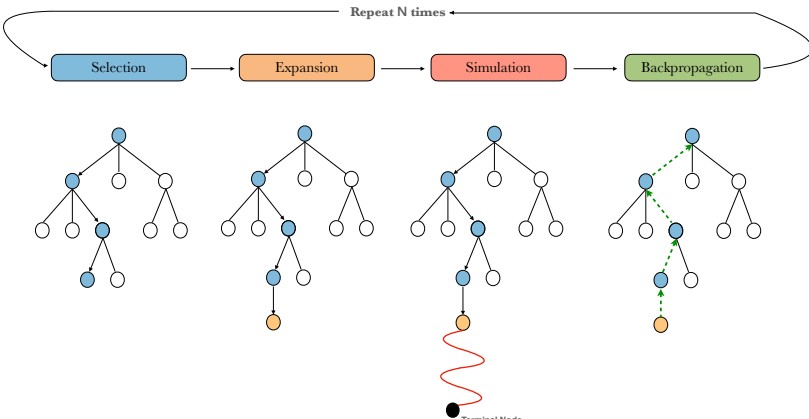

Figure 1: The four major steps of MCTS is depicted in the above figure. These steps are executed sequentially 'N' times.

## 4   Proposed Methodology

### 4.1   Rapid Exploration and Exploitation: REX

The major drawbacks of vanilla MCTS algorithm is that it's computationally expensive and fails to arrive at the solution quickly. To address these issues we propose an accelerated version of MCTS, called REX, by using LLM and In-Context Learning (ICL). LLM's ability to learn from the context and to generate multiple actions at once makes it an ideal candidate for this methodology. In our setup, LLM is the agent. Figure 2a shows the pictorial representation of this algorithm and Figure 3a depicts the flowchart. The four major steps of MCTS can be compressed to two steps in REX as follows:

1. **Selection + Expansion + Simulation:** Rather than progressing step by step from the initial state, taking actions, and moving to the next state until reaching a terminal state, the

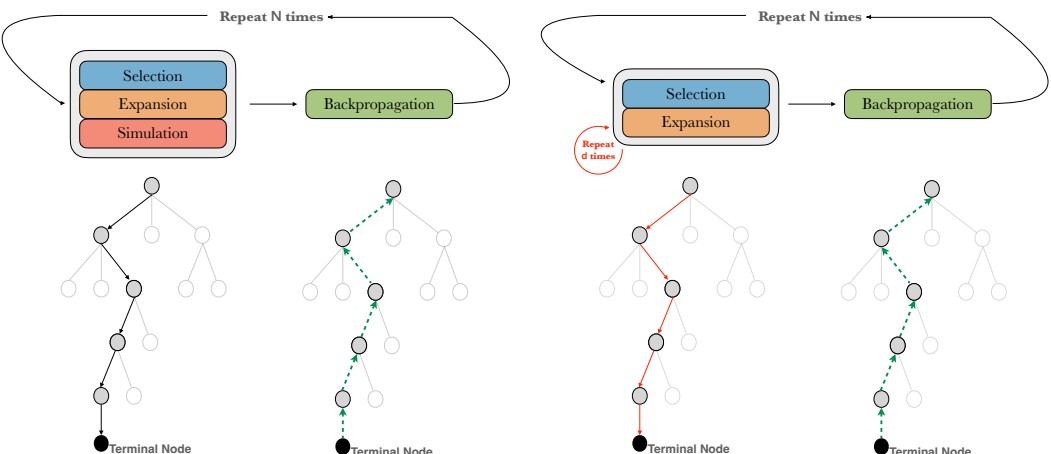

(a) **REX-UCB** The Selection, Expansion, and Simulation steps in MCTS is combined to form one single step in REX.

(b) **REX-UCL** The Selection and Expansion steps are amalgamated into a single step; however, due to the adjustment of logits, all the steps are not simultaneously generated; instead, each step is generated individually.

Figure 2: REX

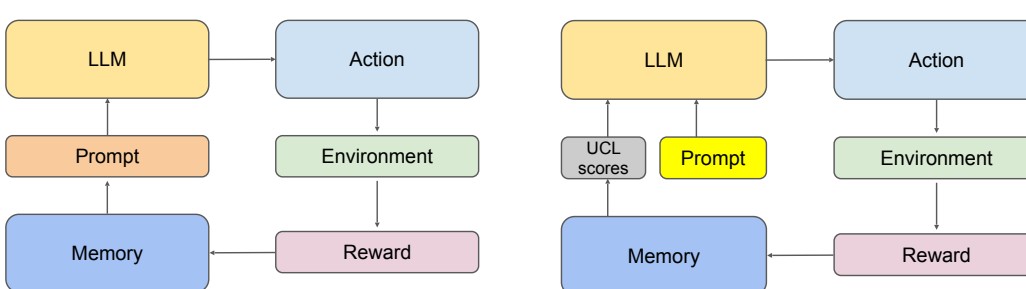

(a) **REX-UCB: Action** denotes the solution produced by the **LLM**, while **Environment** represents the context within which the task is defined. The determination and subsequent updating of the **Reward** within the **Memory** (UCB table) are contingent upon the task's characteristics. Following this, the UCB scores are translated into HIGH or LOW values and are subsequently integrated into the **Prompt** (More details on prompt design is discussed in Appendix B.1)

(b) **REX-UCL: Action** denotes the solution produced by the **LLM**, while **Environment** represents the context within which the task is defined. The determination and subsequent updating of the **Reward** within the **Memory** (UCL table) are contingent upon the task's characteristics. Following this, the UCL scores are directly provided to **LLM** to influence the tokens associated with the desired action sequence (More details on prompt design is discussed in Appendix B.2)

Figure 3: REX Flowchart

new algorithm considers all possible actions simultaneously for each step in the solution. It predicts the entire solution, including the sequence of intermediate steps and the final answer, in one go. This approach removes the need for explicit state transitions and multiple, independent predictions.

2. **Backpropagation:** Based on the correctness of the final solution, the feedback i.e. rewards are propagated back to all the intermediate steps in the solution. This new information is used in the prompt to propose better solution.

MCTS employs the UCB as its selection policy. Correspondingly, the initial iteration of REX, referred to as REX-UCB, also adopts UCB as its selection policy. Depending on the choice of selection policy, various iterations of REX can be formulated. In this research paper, we have

conducted experiments with two additional variants, specifically denoted as REX-$\mathcal{R}$ and REX-UCL, and their detailed descriptions can be found in Sections 4.3 and 4.4, respectively.

## 4.2 ALGORITHM 1: REX-UCB

REX-UCB is the default version of REX, where UCB is used as selection policy.

1. **Approach:** This approach combines the principles of MCTS, CoT, and ICL. In our setup, we define the term `question` or `problem` as the task we aim to solve. We use the term `solution` to encompass both the `sequence-of-intermediate-steps` and the `final-answer`. The `final-answer` represents the ultimate response generated by the model. On the other hand, the `sequence-of-intermediate-steps` refers to the steps that lead to the `final-answer`. We use `action` and `step` interchangeably; in either case it refers to a single step in `sequence-of-intermediate-steps`. we refer `state` to represent the step-index (e.g. step-1, step-2, etc.) in `sequence-of-intermediate-steps`.

2. **Reward Assignment:** After each pass, the language model's solution is evaluated. If the `final-answer` is correct, a reward of +1 is assigned to each step in `sequence-of-intermediate-steps` in the solution, indicating a 'High reward action.' Conversely, if the `final-answer` is incorrect, a reward of 0 is assigned to each step in `sequence-of-intermediate-steps` in the solution, in that pass, representing a 'Low reward action.' Based on the nature of the task and environment, we design reward learning from environment or LLM as follows:

   • **Reward Learning from Environment:** In Blocksworld dataset (discussed in section 5.2), where the expected answer is already known, a reward of +1 is assigned if the `final-answer` matches the target block configuration; otherwise, the reward is 0

   • **Reward Learning from LLM:** In the GSM8K dataset (discussed in section 5.3), where the expected answer is not known, the `final answer`'s correctness is determined by the LLM i.e we provide the `question`, `sequence-of-intermediate-steps`, and `final-answer` from the current pass and query the LLM asking if the solution is correct? If yes, a reward of +1 is assigned else reward is 0

3. **UCB Scoring:** In each pass, the UCB score is calculated for each action taken by the agent. This scoring mechanism encourages the language model to explore different actions while favoring those with higher potential rewards. In the eq. 1, `s` represents the `state`, and `a` represents the `action` taken at `s`. `N(s)` is the number of times agent has produced `s` in it's `solution` and `N(s, a)` is the number of times the agent took an action `a` from a state `s`. `C` is constant to balance the exploration vs. exploitation trade off. `Q̂(s, a)` is the cumulative reward for taking a action `a` at state `s`.

$$\texttt{UCB(s, a)} = \hat{\texttt{Q}}\texttt{(s, a)} + \texttt{C} * \sqrt{\frac{\ln \texttt{N(s)}}{\texttt{N(s, a)}}} \tag{1}$$

4. **Reinforcement Learning and Policy Optimization:** The UCB scores are mapped to `HIGH` or `LOW` tokens and incorporated into the prompts, this approach leverages reinforcement learning techniques to encourage the language model to generate more `HIGH` reward actions and avoid `LOW` reward actions. The agent learns from the rewards associated with each pass and adjusts its policy accordingly. The goal is to optimize the agent's decision-making process over time, leading to improved performance and more accurate solutions.

5. **Selection of Optimal Action Sequence:** After a specified number of passes, the sequence of actions that yields the highest cumulative reward at each stage is selected as the final solution. This selection process ensures that the language model's decision-making is guided by the reinforcement learning framework and prioritizes actions that maximize the cumulative reward, thus promoting the generation of accurate solutions.

More details on the structure of the prompt and a simple example is provided in Appendix B & C.

---

**Algorithm 1** REX-UCB

---

**Require:** Problem $P$, final-answer $Z$, sequence-of-intermediate-steps $H$, action space $A$, state space $S$, action space for state $s$ denoted as $A(s)$ where $s \in S$, number of passes $N$, reward function $Q$: $S$ x $A \to R$, upper confidence bound function $U$: $S$ x $A \to R$, expected reward function $E$: $S$ x $A \to \{\texttt{HIGH}, \texttt{LOW}\}$, expected-answer $X$, reward $\mathcal{R}$

1: AGENT ← Agent( )                                                      */* initialize an agent */*
2: **for** i ← 1 to $N$ **do**
3:    $U(s, a) \leftarrow$ CALCULATEUCB$(s, a), \quad \forall (s, a) \in Q$
4:    **for** $s$ in $S$ **do**
5:        **if** $U(s, a) == \max_{a \epsilon A(s)} U(s, a)$ **then** $E(s, a) \leftarrow \texttt{HIGH}$ **else** $E(s, a) \leftarrow \texttt{LOW}$  */* action with the highest UCB score is mapped to HIGH and other actions are mapped to LOW */*
6:    **end for**
7:    $H, Z \leftarrow$ AGENT.SOLVE$(P, E)$     */* agent predicts intermediate steps and final answer */*
8:    **if** $X$ is available **then**
9:        **if** $Z == X$ **then** $\mathcal{R} \leftarrow +1$ **else** $\mathcal{R} \leftarrow 0$
10:    **else**
11:        $valid\_answer \leftarrow$ AGENT.VALIDATEACTION$(P, H, Z)$
12:        **if** $valid\_answer$ is TRUE **then** $\mathcal{R} \leftarrow +1$ **else** $\mathcal{R} \leftarrow 0$
13:    **end if**
14:    $Q(s, a) \leftarrow Q(s, a) + \mathcal{R}, \quad \forall (s, a) \in H$
15: **end for**

---

## 4.3 ALGORITHM 2: REX-$\mathcal{R}$

REX-$\mathcal{R}$ closely resembles REX-UCB, differ only in the method used to determine the `HIGH` / `LOW` expected rewards. While REX-UCB relies on UCB scores, REX-$\mathcal{R}$ utilizes simple rewards of +1 and 0 for this purpose. This is equivalent to setting `C = 0` in eq. 1. As a result, the exploration aspect becomes void; however, due to our mapping of scores to `HIGH` / `LOW` and our encouragement for the model to exclusively pursue `HIGH` actions, a certain degree of exploration continues to be in effect.

## 4.4 ALGORITHM 3: REX-UCL

In this section we propose a novel paradigm to systematically modify the logits of LLMs called UCL (UCB applied for Logits of LLM). The pictorial representation is shown in Figure 2b and the respective flowchart is shown in figure 3b. This approach is a variant of REX, featuring a distinct modification in its methodology. While REX-UCB employs In-Context Learning, this variation involves adjusting the logits of actions to influence the decision-making process of the LLM instead. In certain cases, language models may generate alternative actions instead of strictly following the instructions in prompt. To address this, UCL adjusts the logits corresponding to tokens associated with the actions, ensuring that the intended actions are consistently chosen.

1. **Approach:** This approach builds upon the foundation of the REX model by introducing novel scoring mechanisms using UCB called UCL. In order to modify the logits for a given query, we adhere to a one-time alteration constraint and therefore execute a query for each `step` in the solution. This approach differs from the previous method, where a single call was made to retrieve the entire solution, as we now make a separate call for each individual `state`. While this approach involves multiple queries to LLMs, akin to the standard MCTS method, it still demonstrates greater speed in comparison to other MCTS-based agents, such as RAP.

2. **Reward Assignment:** Reward assignment for REX-UCL is exactly same as the reward assignment for REX-UCB.

3. **UCL Scoring:** The UCL score is calculated based on eq. 2. This score is used to update the loglikelihoods of tokens corresponding to actions for the current state in the language model. By manipulating the token logits with UCL scores during generation, the agent is compelled to execute the action yielding the highest reward. The constants `B` & `K` control

the extent to which logits of the LLM are offset.

$$\texttt{UCL(s, a)} = \texttt{B} * \ln \frac{\texttt{UCB(s, a)}}{\texttt{K}} \tag{2}$$

4. **Selection of Optimal Action Sequence:** Similar to previous approaches, after a specified number of passes, the sequence of actions that yields the highest cumulative reward at each stage is selected as the final solution.

---

**Algorithm 3** REX-UCL

---

**Require:** Problem $P$, next-action $\hat{a}$, action space $A$, state space $S$, action space for state $s$ denoted as $A(s)$ where $s \in S$, number of passes $N$, reward function $Q$: $S$ x $A \rightarrow R$, upper confidence bound function $U$: $S$ x $A \rightarrow R$, UCL function $L$: $S$ x $A \rightarrow R$, expected-answer $X$, state at depth $d$ denoted as $S_d$, and let $D$ be the maximum permissible depth, constant $B$, constant $K$

1: AGENT ← Agent( )                                    /* initialize an agent */
2: **for** i ← 1 to $N$ **do**
3:    $S_d \leftarrow P$
4:    $trajectory \leftarrow [\,]$
5:    **for** j ← 1 to $D$ **do**
6:       $U(S_d, a) \leftarrow$ CALCULATEUCB$(S_d, a)$, $\forall a \in A(S_d)$
7:       $L(S_d, a) \leftarrow B * \ln \frac{U(S_d,a)}{K}$
8:       **for** $a$ in $A(S_d)$ **do**
9:          $T \leftarrow$ GETTOKENIDS$(a)$                /* get token ids for each token in the action */
10:          $logit\_bias(t) \leftarrow L(S_d, a)$, $\forall t \in T$
11:       **end for**
12:       $\hat{a} \leftarrow$ AGENT.SOLVE$(P, logit\_bias)$
13:       **if** $X$ is available **then**
14:          **if** $\hat{a} == X$ **then** $\mathcal{R} \leftarrow +1$ **else** $\mathcal{R} \leftarrow 0$
15:       **else**
16:          $valid\_answer \leftarrow$ AGENT.VALIDATEACTION$(P, S_d, \hat{a})$
17:          **if** $valid\_answer$ is TRUE **then** $\mathcal{R} \leftarrow +1$ **else** $\mathcal{R} \leftarrow 0$
18:       **end if**
19:       $trajectory \leftarrow trajectory + (S_d, \hat{a})$        /* append state & action to trajectory */
20:       $Q(s, a) \leftarrow Q(s, a) + \mathcal{R}$, $\forall (s, a) \in trajectory$
21:       **Update:** $S_d \leftarrow S_d + \hat{a}$                /* append action to state */
22:    **end for**
23: **end for**

---

## 5  EXPERIMENTS & DISCUSSION

### 5.1  BASELINE

1. **CoT:** The Multi-pass CoT (Wei et al., 2022) builds upon the idea that involves designing prompts that include a series of step-by-step examples, ultimately leading to the final solution. A successful outcome is achieved if at least one of these queries yields a correct answer.

2. **Reflexion:** Reflexion (Shinn et al., 2023) framework leverages LLMs to strategically plan and execute a task. Reflects on it's decision making and then improvises.

3. **RAP:** RAP (Hao et al., 2023) employs a conventional Monte Carlo Tree Search approach to address the assigned task, reutilizing the LLM in the dual roles of a world model and a reasoning agent. Our primary focus here is the comparative analysis of time complexity.

For our experiments we have used Blocksworld dataset and GSM8K dataset. More details on these two datasets are presented in the following sections.

## 5.2 BLOCKSWORLD

The Blocksworld dataset (Valmeekam et al., 2023) represents a planning problem that involves the arrangement of blocks with varying colors in a predetermined configuration. Each instance of the dataset provides an initial block configuration and the desired final configuration. The term 'block configuration' refers to the specific arrangement of the blocks, where each block can be positioned either on top of another block, on a table surface, or held in hand, but not all options are available simultaneously. The blocks are uniquely identified by their respective colors, such as red, blue, and so on. The Blocksworld dataset is divided into 3 subcategories (2, 4, 6 steps) based on the number of steps required to transform the block configuration from initial to final.

To transform the block configuration from initial to final, the model is expected to propose sequence of steps/actions. In blocksworld, there are only four major actions - STACK, UNSTACK, PICK and PUT. Table 1 illustrates the performance of the methodologies proposed in this study. Each experiment is run for 10 iterations/passes and chatGPT (model: gpt-3.5-turbo) is used for all the experiments. The best score is in bold and the second best score is underlined. In order to assess the efficacy of the proposed algorithm more accurately, we compare the performance of the different algorithms with temperature (T) of LLMs is set to 0.0. Setting a higher temperature value can obscure the impact of the proposed approaches, making it difficult to evaluate their effectiveness.

## 5.3 GSM8K

The GSM8K dataset (Cobbe et al., 2021) comprises a collection of 8.5k grade school math word problems of exceptional quality. For our experiment we have used GSM8K test set which has roughly 1.3k samples. Each problem within this dataset typically requires a solution involving a sequence of elementary calculations, utilizing fundamental arithmetic operations such as addition, subtraction, multiplication, and division (+, -, ×, ÷). The number of steps required to solve each problem falls within the range of 2 to 8 steps. The performance of the proposed methodologies are presented in Table 1. Temperatures is set to 0.0 for all the methods.

Table 1: Accuracy & Time complexity of various models

| Architecture | Blocksworld | | | GSM8K-test | Time complexity |
| | 2-step (size=30) | 4-step (size=56) | 6-step (size=114) | 2-to-8-steps (size=1319) | |
| --- | --- | --- | --- | --- | --- |
| CoT | 40% | 17.85% | 8.77% | 80.81% | n |
| Reflexion (+ CoT) | 41.67% | 41.96% | **29.82%** | 88.85% | 3*n |
| RAP | - | - | - | - | n*m*d |
| REX-$\mathcal{R}$ | 53.33% | 37.5% | 14.91% | 81.34% | n |
| REX-UCB | **80%** | 39.28% | 25.43% | 82.03% | n |
| REX-UCL | 60% | **44.64%** | 20.17% | **90.44%** | n*d |

## 5.4 REX: ACCURACY, SPEED, AND LIMITATIONS

**Accuracy:** The accuracy of various methods presented in Table 1 unequivocally demonstrates REX's superior performance in comparison to alternative techniques such as CoT and Reflexion, particularly on the Blocksworld (2 & 4 step) and GSM8K datasets. While it is noteworthy that no single REX variation consistently outperforms every algorithm across all datasets, the average performance of REX consistently exhibits highly promising results. RAP is not compatible with OpenAI APIs; therefore, we have exclusively focused on evaluating the time complexity of RAP, without delving into its accuracy.

**Time complexity:** In this context, we define time complexity as the count of queries made to the LLM in order to accomplish a given task. We introduce three pivotal variables: 'n,' which signifies the number of iterations or passes; 'd,' denoting the depth limit or the maximum count of intermediate steps; and 'm,' indicating the quantity of potential actions generated at each state. The last column in Table 1 presents the time complexities of various methods. Our approach not only

excels in computational efficiency but also offers heightened flexibility. Notably, the integration of REX enhancements is viable within any LLM, including widely adopted APIs such as OpenAI, even in cases where access to the underlying logits is restricted. This adaptability underscores REX as an exceptionally versatile and pragmatic choice suitable for a wide spectrum of applications. In stark contrast, RAP requires access to logit values for tokens associated with all actions to estimate rewards, rendering it less practical for real-world scenarios.

**Limitations:** Numerous uncharted territories in this field await exploration, including but not limited to context length limitations, in-context learning intricacies, and the scope of generalization capabilities. Herein, we elucidate key facets of REX's performance and potential limitations: (1) REX impressively demonstrates formidable accuracy and efficient time complexity. However, its efficacy is tethered to context length, owing to its reliance on In-Context Learning (ICL); (2) Effective utilization of REX necessitates a skillful reconfiguration of the prompt, tailored meticulously for each specific task domain; (3) The C, B & K parameters in UCB and UCL require domain-specific adjustments. Furthermore, harnessing the power of REX-UCL requires a nuanced manipulation of the logit values within the LLMs.

## 5.5 REX: UCB FOR EFFECTIVE EXPLORATION AND EXPLOITATION

From Table 1 it is clear that UCB based feedback outperforms simple reward $\mathcal{R}$ (UCB with C=0) based feedback in Planning (Blocksworld) and Mathematical Reasoning (GSM8K) datasets. An example problem solved by REX-$\mathcal{R}$ and REX-UCB is presented in the Appendix C.

The examination detailed in Table 2, which delineates the mean count of distinct actions at each intermediate solution stage, elucidates that the REX-UCB strategy fosters a more pronounced inclination for action exploration at each juncture when compared with the REX-$\mathcal{R}$ approach. This strategic prioritization of exploration within the action space is posited to enhance the agent's ability to uncover innovative problem-solving pathways, consequently augmenting the success rate. REX, through its incorporation of UCB-based feedback, strikes a delicate equilibrium between exploration and exploitation, ultimately yielding enhanced performance.

Table 2: Avg. number of unique actions for each intermediate step after 10 iterations

| Architecture | Blocksworld | | |
| --- | --- | --- | --- |
| | 2-step (size=30) | 4-step (size=56) | 6-step (size=114) |
| REX-$\mathcal{R}$ | 1.63 | 2.32 | 2.34 |
| REX-UCB | 3.61 | 3.89 | 4.24 |

## 6 CONCLUSION

In conclusion, this paper presented REX and it's variations, aimed at enhancing the performance of AI agents. These techniques effectively strike a balance between exploration and exploitation, which is crucial for successful problem-solving. Through extensive evaluations on Planning and Mathematical Reasoning datasets, we have demonstrated the superiority of REX. The strengths of REX lie in its simplicity, efficiency, flexibility, and speed, making it a compelling choice for real-world applications. By incorporating UCB-based formula and strategically allowing the model to predict multiple steps at once, our techniques provide a robust framework for optimizing AI agents, opening doors for advancements in various domains that require large-scale action modeling.

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

# Appendix

## A    Synthesizing Large Language Models with Reinforcement Learning for Enhanced AI agents

Ideally, an AI agent is anticipated to autonomously execute designated tasks. To achieve this, the AI agent necessitates unfettered access to an assemblage of information, factual details, rules, and pertinent data. Furthermore, the agent must possess the faculty to engage in rational decision-making processes and subsequently effectuate task completion. In the realm of advanced AI agents, an additional facet is the imperative capacity to engage with the surroundings, assimilate insights from errors, and progressively refine their acumen in the sphere of decision making. To facilitate the realization of these functionalities, the amalgamation of Large Language Models (LLMs) and Reinforcement Learning (RL) emerges as a highly auspicious trajectory.

### A.1    Why LLMs are a good choice?

LLMs are deep neural networks that have been trained on vast amounts of text data to predict the next word in a sequence (Vaswani et al., 2017; Brown et al., 2020; Devlin et al., 2019; OpenAI, 2023). They learn the statistical patterns and semantic relationships between words, allowing them to generate contextually relevant text. LLMs are good at logical reasoning and Decision making to some extent.

### A.2    Some major limitations

Large Language Models (LLMs) possess several limitations that stem from their lack of physical embodiment and sensory perception. LLMs heavily rely on precise instructions, grappling with ambiguity and misinterpreting vague commands, potentially yielding erroneous outcomes. Their knowledge, drawn from vast textual data, often lacks common sense reasoning and contextual understanding beyond explicit mentions. This results in technically accurate yet unexpected responses. Moreover, they struggle to provide causal reasoning, hindering the capacity to offer justifications or detailed explanations for decisions. Lastly, LLMs struggle to generalize to novel situations, functioning well in familiar tasks but producing unreliable outcomes when exposed to uncharted inputs.

### A.3    Reinforcement Learning (To overcome these limitations)

Reinforcement Learning (RL) empowers Language Model agents (LLMs) to learn optimal decision-making by interacting with an environment, receiving input text, and generating responses or actions. This process involves balancing exploration and exploitation, as the LLM agent gradually discovers effective strategies through experimentation and then leverages that knowledge to maximize rewards. Key to RL's success is defining a reward signal that guides learning; for LLMs, reward shaping provides feedback on response quality, encouraging desired behavior while penalizing errors. RL also facilitates training in simulated environments, enabling LLMs to acquire skills applicable to real-world scenarios. At the core of RL is the optimization of the agent's policy—typically a neural network—dictating actions in response to input text. One such prominent algorithm is Monte Carlo Tree Search(MCTS).

An example instance from blocksworld dataset(2-step) is presented in Appendix. Figure 4 shows the solution provided by REX-$\mathcal{R}$ and Figure 5 shows the solution provided by REX-UCB. This is an example which shows how REX-UCB was able to solve a problem which REX-$\mathcal{R}$ wasn't able to solve. Clearly, encouraging the model to pick a HIGH reward action based on UCB score has helped the model to solve the given problem.

The REX-$\mathcal{R}$ approach involves providing the model with specific information and instructions to solve a problem during each pass. The prompt includes details about the problem, instructions, action space, and three examples of solved problems (3-shot examples). Additionally, feedback from previous passes is given to the model in each pass. In the Simple-Reward($\mathcal{R}$) setting, if the action sequence leads to the correct answer, a 'HIGH' reward is assigned to each action or step in the solution. Conversely, if the action sequence does not lead to the correct answer, a 'LOW' reward is

assigned. This reward information is included in the prompt for the subsequent pass. It's important to note that unless the action sequence results in a correct answer, none of the actions or steps will receive a 'HIGH' reward. Consequently, the model is encouraged to propose new actions in order to improve its performance.

The REX-UCB solution is depicted in Figure 5. Similar to the REX-$\mathcal{R}$ approach, the REX-UCB prompt incorporates comprehensive information about the problem, including problem details, instructions, action space, and three examples of solved problems (referred to as 3-shot examples). Moreover, feedback from previous passes is incorporated into the model at each pass. We compute the Upper Confidence Bound (UCB) score for each action within the solution during each pass. Subsequently, we assign the action associated with the highest UCB score as a 'HIGH' reward, while the remaining actions are designated as 'LOW' reward. This UCB scoring mechanism ensures an effective selection process, by striking a good balance between exploration and exploitation, for identifying the most promising action to be executed, optimizing the model's performance within the REX-UCB framework.

# B  PROMPT STRUCTURE

## B.1  REX-UCB

The prompt for LLM has three major parts: (1) **Task Instruction** The set of specific directions, guidelines, or information provided to LLM to complete a particular task or achieve a specific objective. These instructions typically outline what needs to be done, how it should be done, and any relevant details or requirements to ensure the successful execution of the task. Task instructions can vary widely depending on the context and nature of the task, and they play a crucial role in guiding and facilitating tasks or assignments; (2) **Few-Shot examples** The model is expected to generalize from this limited set of examples to perform tasks or make predictions on new, unseen data; (3) **State-Action-Feedback** Expected reward, based on UCB scores, for each action in every state.

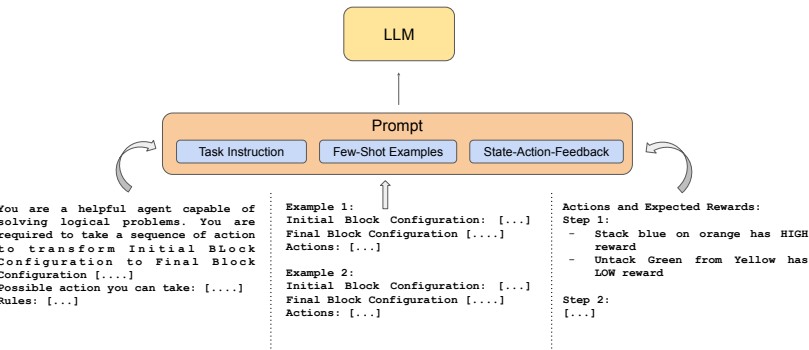

Figure 4: Prmpt for REX-UCB

## B.2  REX-UCL

The **Task Instruction** and **Few-Shot Examples** components are the same as in REX-UCB. **State-Action-Feedback** component is removed and instead UCL scores are provided to LLM to adjust the logit values.

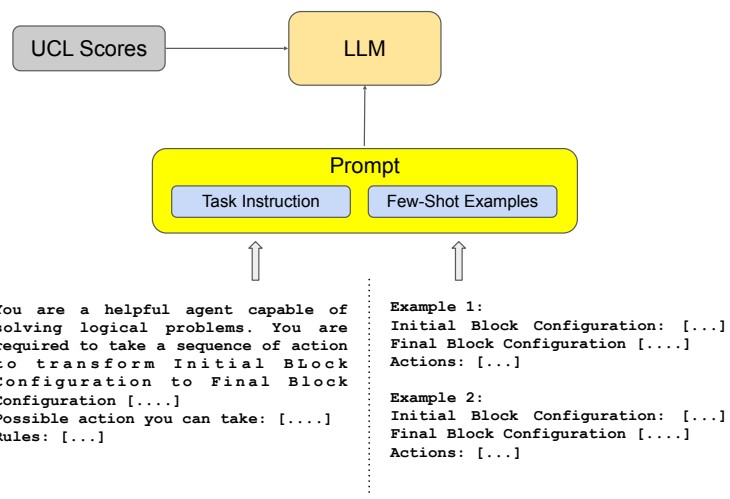

Figure 5: REX-UCL

**Initial Block Configuration**

Blue block is on the table. Red block is on top of Blue block and is free. Yellow block is on table. Orange block is on top of Yellow block and is free

**Final Block Configuration**

Blue block is on the table. Red block is on top of Blue block. Yellow block is on table and is free. Orange block is on top of Red block and is free

**Ground Truth**

**Step 1:** Unstack Orange block from on top of Yellow block
**Step 2:** Stack Orange block on top of Red block

Figure 6: Sample problem from Blocksworld

## C   SAMPLE SOLUTION PROVIDED BY REX-$\mathcal{R}$ AND REX-UCB

An example from the 2-step block configuration is shown in Figure 6. Initial Block Configuration represents how the colored block are are arranged initially and Final Block Configuration shows the expected block arrangement. Ground Truth is the sequence of actions that needs to be taken to transform Block Configuration from Initial to Final. Figure 7 shows the solution provided by REX-$\mathcal{R}$ and Figure 8 shows the solution provided by REX-UCB. After 10 passes, REX-$\mathcal{R}$ was unable to solve the task. On the other hand, REX-UCL solved the problem in 7th pass.

| | | | | |
|---|---|---|---|---|
| **Simple-Reward-CoT Solution:** | | | | |
| **Model Input** | **Pass-1 Prompt:**

[. . .] | **Pass-2 Prompt:**

[. . .]

Actions and Expected Rewards:

Step 1:
Pick Yellow block has LOW reward

Step 2:
Unstack Orange block from on top of Yellow block has LOW reward | . . . | **Pass-2 Prompt:**

[. . .]

Actions and Expected Rewards:

Step 1:
Pick Red block has LOW reward
Pick Orange block has LOW reward
....

Step 2:
Unstack Orange block from on top of Yellow block has LOW reward
Stack Red block on top of Blue block has LOW reward
..... |
| **Model Output** | **Pass-1 Action Sequence:**
**Step 1:** Pick Yellow block
**Step 2:** Unstack Orange block from on top of Yellow block | **Pass-2 Action Sequence:**
**Step 1:** Pick Red block
**Step 2:** Stack Red block on top of Blue block | . . . | **Pass-10 Action Sequence:**
**Step 1:** Unstack Red block from on top of Blue block
**Step 2:** Put Red block |

Figure 7: REX-$\mathcal{R}$ based Solution

| | | | | |
|---|---|---|---|---|
| **UCB-CoT Solution:** | | | | |
| **Model Input** | **Pass-1 Prompt:**

[. . .] | **Pass-2 Prompt:**

[. . .]

Actions and Expected Rewards:

Step 1:
- Unstack Orange block from on top of Yellow block has LOW reward
- Stack based action has HIGH reward
- Pick based action has HIGH reward
- Put based action has HIGH reward

Step 2:
- Pick Red block has LOW reward
- Stack based action has HIGH reward
- Unstack based action has HIGH reward
- Put based action has HIGH reward | . . . | **Pass-7 Prompt:**

[. . .]

Actions and Expected Rewards:

Step 1:
- Unstack Orange block from on top of Yellow block has LOW reward
- Stack Yellow block on Red block has LOW reward
- Pick Yellow block has LOW reward
...
- Put based action has HIGH reward

Step 2:
- Pick up Red block has LOW reward
- Unstack Orange on top of Yellow has LOW reward
....
- Stack Orange block on top of Red block has HIGH reward |
| **Model Output** | **Pass-1 Model's Output:**

**Step 1:** Unstack Orange block from on top of Yellow block
**Step 2:** Pick Red block | **Pass-2 Model's Output:**

**Step 1:** Pick Red block
**Step 2:** Stack Red block on top of Blue block | . . . | **Pass-7 Model's Output:**

**Step 1:** Unstack Orange block from on top of Yellow block
**Step 2:** Stack Orange block on top of Red block |

Figure 8: REX-UCB based Solution

## D  MORE DETAILS ON THE 'AGENT.XXXX()' METHODS USED IN THE ALGORITHMS

1. **CALCULATE_UCB():** This method simply calculates the UCB score for every (state, action) using the formula presented in eq. 1

   - **Input:** state $s$, action $a$
   - **Output:** a scalar value

2. **AGENT.SOLVE():** This method represents the inference call made to the LLM. In our case, this is exactly same as OAI's chat.completion() API. It takes in the prompt and the expected-reward-values for each (state, action).

   - **Input:** prompt/problem with instructions $P$, expected reward function $E$
   - **Output:** It produces two values in a tuple: $H$, which represents a sequence of intermediate steps, and $Z$, which represents the final answer.

3. **AGENT.VALIDATE_ACTION():** This method represents the inference call made to the LLM. In our case, this is exactly same as OAI's chat.completion() API. It takes in the prompt, the sequence-of-intermediate-steps, and the final answer proposed by the model i.e. output of AGENT.SOLVE()

   - **Input:** prompt/problem with instructions $P$, sequence of intermediate steps $H$, final answer $Z$
   - **Output:** It returns a Boolean values i.e True or False

