# OpenReview forum: "REX: Rapid Exploration and eXploitation for AI agents"
_ICLR.cc/2024/Conference — Submitted to ICLR 2024_

### Official Review · Reviewer_1Dq3 · 2023-10-16

**Soundness:** 2 fair
**Presentation:** 3 good
**Contribution:** 2 fair
**Rating:** 5
**Confidence:** 2

**Summary:**

The paper motivates their method by identifying current ICL-agents lack: 1) systematic rewards incorporation, 2) Exploration-Exploitation trade-off, 3) Accuracy and latency. They introduced a method called Rapid Exploration and eXploitation for AI agents (REX) to address the aforementioned issues.

In particular, REX uses MCTS as the base search algorithm to

**Strengths:**

- The paper explores a very relevant topic that is less looked at, which is the exploration problem in reasoning tasks.
- The paper is easy to read, explains their method well
- The motivation of exploration and exploitation is clear and is a topic that needs attention in LLM-based reasoning/agents

**Weaknesses:**

- The evaluation is somewhat lacking, only evaluated on blocksworld and GSM8K, and ablations were missing (only statistics of # of actions were given), no intuition on how to tune the hyperparameters B,K,C
- Baselines is missing, see question 1
- some implementation details are missing, see question 2
- needs more intuition/analytics for how REX works, which one is better in what situation, when to use it, see question 3 & 4

**Questions:**

How come RAP scores are missing for every evaluated environment? It is mentioned RAP is not compatible with OAI APIs, maybe supply more results with open-source models?

how was each action generated? If im not mistaken they are each reasoning steps. And if temperature is 0, how come there are variations to the steps generated when prompted twice? is it caused by using previous history of steps in the UCB prompt (State-Action-Feedback in figure 4)?

Do you have some intuition on when is REX-UCL better than REX-UCB and vice-versa? (Paper only included REX-UCL is better at exploration than REX-R)

How come when more steps (6) is required, method is worse than Reflexion? Shouldn't it be the opposite, since REX employs search trees. Do you have any intuition?

---

> ### Author Response · Authors · 2023-11-21
>
> Thank you for your time and valuable feedback! We have tried to address your concerns below:
>
> ---
> Weakness:
>
> ---
> **The evaluation is somewhat lacking, only evaluated on blocksworld and GSM8K, and ablations were missing (only statistics of # of actions were given), no intuition on how to tune the hyperparameters B,K,C**
> * The primary rationale behind selecting Blocksworld and GSM8K is rooted in their suitability for evaluating distinct facets: Blocksworld provides an ideal dataset for assessing the planning capability of LLM and its respective technique, while GSM8K offers a suitable dataset for evaluating problem-solving abilities. We consider these two categories of datasets, focusing on Planning and Problem Solving, as strong benchmarks for our evaluation.
> * B, K, C values we set based on the grid search technique. We set a reasonable range for all the parameters - $1 \leq C \leq 10000$
> , $1 \leq B \leq 1000$, $1 \leq K \leq 1000$ and ran a grid search to approximate the values.
>
>
> ---
> Questions:
>
> ---
>
> **How come RAP scores are missing for every evaluated environment? It is mentioned RAP is not compatible with OAI APIs, maybe supply more results with open-source models?**\
> We added RAP to demonstrate its slower performance compared to REX in problem-solving. Reflexion and CoT were used as other advanced techniques for performance comparison.
>
> **how was each action generated? If im not mistaken they are each reasoning steps. And if temperature is 0, how come there are variations to the steps generated when prompted twice? is it caused by using previous history of steps in the UCB prompt (State-Action-Feedback in figure 4)?**
> * The LLM receives input comprising details about the problem requiring a solution, a set of few-shot examples, and potential actions derived from previous interactions, each categorized as HIGH or LOW (as outlined in Algorithm 1 in the paper). The LLM is then prompted to select an action with a higher likelihood of success at each stage. Further elaboration on the specific prompt structure and action generation is provided in Section B of the Appendix, while Section C contains illustrative examples.
> * Based on the trajectory and the final answer generated by the LLM, rewards are computed following the procedure outlined in Algorithm 1 in the paper. These rewards are utilized to calculate UCB scores (equation 1), which are further categorized as HIGH or LOW. Subsequently, these categorized scores are incorporated into the prompt for the succeeding iteration. In essence, the Agent suggests a solution, receives feedback, and then proposes a different solution in the subsequent iteration based on this received feedback. In our context, the Agent is represented by the LLM, and each prompt iteration contains updated information on the effectiveness of the (state, action) pair. Therefore, setting the temperature (t) to 0 should not inhibit the Agent from suggesting alternative solutions. More details on this is available in Section B & C of the Appendix.
>
> **Do you have some intuition on when is REX-UCL better than REX-UCB and vice-versa? (Paper only included REX-UCL is better at exploration than REX-R)**
> * Accuracy:
>     * REX-UCL demonstrates strong performance in mathematical reasoning scenarios, particularly in cases with a significantly expansive action space. For instance, in GSM8K where the number of operands can be extensive, REX-UCL effectively narrows down the action space by effectively suppressing the respective logits.
>     * REX-UCB tends to perform well in planning scenarios, where the action space is bounded.
> * Speed:
>     * REX-UCB is faster than REX-UCL, below is the number of API calls made by these two techniques for Blocksworld and GSM8K datasets:
> | Method	| Blocksworld |	GSM8K |
> |-----------|-------------|-----------|
> |REX-UCB|	10	|10|
> |REX-UCL|	60	|80|
> (For Blocksworld, depth(d)=6, number of passes(n)=10, number of potential actions(m)=4; For GSM8K, depth(d)=8, number of passes(n)=10, number of potential actions(m)=4)
>
> **How come when more steps (6) is required, method is worse than Reflexion? Shouldn't it be the opposite, since REX employs search trees. Do you have any intuition?**\
> Reflexion utilizes natural language feedback, whereas REX employs a simpler form of feedback, either HIGH/LOW or a scalar value. We posit that in specific scenarios, natural language feedback may convey more valuable information compared to our proposed feedback system. Consequently, Reflexion outperforms REX in the 6-step subset. Our approach is particularly advantageous in real-world situations where the true nature of an action remains ambiguous, such as when selecting an API. In such cases, we believe our strategy holds an advantage. Results in Table 1 are averaged over 10 runs/iterations. While 10 iterations suffice for a 2-step problem to find a solution, arriving at the correct solution quickly may be more challenging for a 6-step problem.

---

> > ### Comment · Reviewer_1Dq3 · 2023-11-22
> >
> > Thank you for your clarifications, I do not have any additional ones.

---

### Official Review · Reviewer_b79r · 2023-10-23

**Soundness:** 2 fair
**Presentation:** 2 fair
**Contribution:** 2 fair
**Rating:** 3
**Confidence:** 4

**Summary:**

The paper addresses the problem of improving LLM-based reasoning techniques through algorithms that wrap the core LLM inferences. To this end, the paper investigates ways to use Monte Carlo Tree Search with LLM reasoning, by tracking the success of different reasoning traces and using this to guide subsequent reasoning attempts. Two approaches are proposed:
1) Planning full trajectories and using upper confidence bound (UCB) scores to guide the selection of actions from the sequence
	- Uses in-context learning as the mechanism to update model action reward estimates
	- Feedback is provided to the LLM by labeling prompt actions as high or low reward
2) Planning at the level of individual actions and using UCB to reweight predicted token logits for different actions

Three algorithms are proposed (a variant of the first approach uses only raw rewards, rather than UCB scores) and evaluated on a block arrangement task and grade school mathematics problems. The proposed algorithms are compared with two baselines from the literature and found to have higher accuracy in some tasks. The new algorithm is shown to have favorable time complexity compared to three baselines.

**Strengths:**

# originality
Combining search techniques with LLMs is an emerging area. MCTS is only recently being studied in this context, making the work original in that respect.

# quality
The proposed algorithms are evaluated on two different domains with varying size and reasoning requirements. Both accuracy and time complexity are compared.

# clarity
The figures and algorithm listings help clarify the core algorithms.

# significance
Enabling LLMs to incorporate feedback into reasoning is a crucial capability for amortizing expensive inference and coping with partially unknown application domains. This is an important problem and new techniques to address this problem - like using MCTS to structure search and track action quality and uncertainty - are important. The work aims to leverage MCTS while reducing the number of LLM invocations needed compared to prior work (RAP).

**Weaknesses:**

# originality
The MCTS formulation is similar to RAP in some ways, but differs in many key details, particularly around how the world modeling is done, where RAP explicitly models the world but REX uses the implicit model of the LLM. Augmenting LLMs with search techniques is not itself novel, but this is not a substantial lack of originality per se either.

# quality
Evaluation results only weakly support the REX models. The performance improvements are primarily for (some) small Blocksworld domains, with comparable performance on GSM8K. This makes it unclear to what degree the algorithm is an improvement over existing methods.

Benchmarking is only done on relatively simple example tasks, making it difficult to assess how well these results translate to real world use cases.

Evaluation does not match techniques on the number of tokens they generate or evaluation calls they make. This makes fair comparisons impossible, as the different methods are afforded different computational budgets for the same objective.

# clarity
The text can be hard to follow. Partially this is due to framing the technique as reinforcement learning, despite the lack of a policy being learned (though presumably the MCTS count tables could be preserved and treated as a learned policy of sorts). The core algorithm description was difficult to follow and the main figures did not support understanding this algorithm.

# significance
The results do not show substantial improvements over prior efforts, limiting the significance attributable to the technical novelties.

**Questions:**

- Figure 2
	- Consider adjusting the visuals here to make the differences clearer. Perhaps by showing where LLM evaluations occur along the search tree to make it clearer that REX-UCL is evaluating the LLM for each step of the depth of the tree.
	- Figure 3 was helpful, but I could not understand what Figure 2 was meant to illustrate by highlighting some lines in the rollout as red.
- Table 1
	- Is it possible to provide uncertainty estimates on the scores provided? (For example, using multiple seeds for different runs of the algorithms)
	- What does "size" mean in this table?
	- What is the intuition for why the REX methods do worse in the largest Blocksworld scenario? They show strong advantages in the 2-step scenario (80% vs 41.7%), but their advantage shrinks in the 4-step scenario (44.6% vs 42.0%) and becomes a deficit in the 6-step scenario (25.4% vs 30.0%).
	- Is there other evidence for the strength of REX on GSM8k-test? The results look very close to CoT for REX-UCB and to Reflexion for REX-UCL. Could the results be subdivided by number of steps required?
		- Note that Reflexion requires `3n` time complexity, while REX-UCL requires `dn`, so on a 3-step problem the two are the same (if I understood correctly), while on a 4 or more step problem REX-UCL will be more computationally demanding unless search depth is limited.
	- What search depth(s) (`d`) were used for the experiments?
- Section 5.4
	- Could the results in table 1 be replicated with a model that exposes logits and compared?
		- These additional results would strengthen the claims about the core technique.
		- This would also enable a comparison to RAP.
	- How were `C`, `B`, and `K` selected for the experiments?
- General
	- Consider computing the results with (approximately) matching compute budgets for the methods being compared. Ideally this would be a matching number of generated tokens. A more coarse comparison would be matching the number of inference calls made. Regardless, some baseline equivalence should be established for the compute budget to allow fair comparison of the accuracy.

---

> ### Author Response · Authors · 2023-11-21
>
> Thank you for your time and valuable feedback! We have tried to address your concerns below:
>
> ---
> Questions:
>
> ---
>
> **- Consider adjusting the visuals here to make the differences clearer. Perhaps by showing where LLM evaluations occur along the search tree to make it clearer that REX-UCL is evaluating the LLM for each step of the depth of the tree.**\
> **- Figure 3 was helpful, but I could not understand what Figure 2 was meant to illustrate by highlighting some lines in the rollout as red.**\
> The primary purpose of Figure 2 is to illustrate the transformation of the classical MCTS technique into various REX variations. Figure 2(b) specifically details the rollout strategy for REX-UCL. The phrase 'Repeat d times,' highlighted in red in Figure 2(b), signifies invoking the LLM 'd' times. As each invocation of LLM triggers the simultaneous execution of the Selection and Expansion steps within the REX-UCL strategy, the loop is situated under these blocks. Additionally, the red arrow marks in Figure 2(b) represent the step-by-step invocation of LLM. Further elaboration on the LLM input, specifically the prompt structures for REX-UCB and REX-UCL, is provided in section B of the Appendix.
>
> **Is it possible to provide uncertainty estimates on the scores provided? (For example, using multiple seeds for different runs of the algorithms)**\
> We have set the temperature to 0.0 to ensure that it does not obscure the effectiveness of the proposed approach and to facilitate a stable comparison. The following stats were derived from 5 runs each with gpt-3.5-turbo-0301 LLM. We applied REX-UCB and REX-UCL to the Blocksworld dataset to demonstrate the observable level of uncertainty. For the results reported in the paper, the same experimental setup was used for other techniques (CoT, Reflexion) as well. The mean(μ) and standard deviation(σ) of number of correct answers across 5 runs is presented below:
>
> | Method | Blocksworld (2 step) | Blocksworld (4 step) | Blocksworld (6 step) |
> |----------|-----------|----------|-----------|
> |  | No of questions = 30 | No of questions = 56 | No of questions = 114 |
> | REX-UCB | μ=24, σ=0.4 | μ=21.6, σ=0.89 | μ=28.8 σ=1.63 |
> | REX-UCL | μ=18, σ=0.4 | μ=24.8, σ=0.48 | μ=22.8, σ=0.47 |
>
> **Note:** In the paper, the results reported are average over 10 runs.
>
> **What is the intuition for why the REX methods do worse in the largest Blocksworld scenario?**\
> As the complexity of a problem increases (measured in terms of the number of steps required to solve it), the permutations of action sequences, i.e., trajectories, also increase. For instance, in the 6-step Blocksworld problem, which comprises 6 steps involving 4 actions (pick, put, stack, unstack) and several blocks, there are approximately 6 * 4 * 4 (= 96) potential actions to consider simultaneously. This potentially contributes to a decline in the model's performance. Moreover, we conducted all experiments with 10 runs/passes. Increasing the number of runs for more complex problems could potentially enhance performance.
>
> **Is there other evidence for the strength of REX on GSM8k-test?**\
> As already stated, the experimental results were obtained by averaging the results across 10 random runs. This supports the robust and consistent superiority of REX over Reflexion and CoT. As you have pointed out, REX has superior speed when d<=3.
>
> **What search depth(s) (d) were used for the experiments?**\
> We have set d=6 for Blocksworld and d=8 for GSM8K.
>
> **Could the results in table 1 be replicated with a model that exposes logits and compared?**\
> (Currently working on it. Will share the results soon.)
>
> **How were C, B, and K selected for the experiments?**
>  The above values we set based on the grid search technique. We set a reasonable range for all the parameters - $1 \leq C \leq 10000$, $1 \leq B \leq 1000$, $1 \leq K \leq 1000$ and ran a grid search to approximate the values.
>
> **Consider computing the results with (approximately) matching compute budgets for the methods being compared. Ideally this would be a matching number of generated tokens. A more coarse comparison would be matching the number of inference calls made. Regardless, some baseline equivalence should be established for the compute budget to allow fair comparison of the accuracy.**\
> The time complexity presented in Table 1 roughly addresses this concern. Below is the number of inference calls made by each of the techniques for every task/problem in Blocksworld and GSM8K datasets:
> (For Blocksworld, depth(d)=6, number of passes(n)=10, number of potential actions(m)=4; For GSM8K, depth(d)=8, number of passes(n)=10, number of potential actions(m)=4)
> | Method | Blocksworld | GSM8K |
> |----------|-----------|----------|
> | CoT | 10 | 10 |
> | Reflexion | 30 | 30 |
> | RAP | 240 | 320 |
> | REX-R | 10 | 10 |
> | REX-UCB | 10 | 10 |
> | REX-UCL | 60 | 80 |

---

> > ### Comment · Reviewer_b79r · 2023-11-22
> >
> > Thank you for the responses! The answers clarified several areas I did not understand.
> >
> > > Below is the number of inference calls made by each of the techniques for every task/problem in Blocksworld and GSM8K datasets
> >
> > It looks like REX-UCL benefits from substantially more compute than CoT or REX-R & REX-UCB. What do the results in Table 1 look like if the other methods are allocated 60 / 80 inference calls?

---

> > > ### Author Response · Authors · 2023-11-22
> > >
> > > That's definitely an interesting point. However, it's crucial to consider the task variations across CoT, REX-R, REX-UCB, and REX-UCL. In CoT, REX-R, and REX-UCB, the agent is tasked with predicting the entire solution—comprising both intermediate steps and the final answer—in a single attempt. On the other hand, REX-UCL requires the agent to predict either a single intermediate step or the final answer within one attempt. As a result, although REX-UCL necessitates more inference calls, the total number of generated tokens remains approximately consistent. Consequently, we believe that the computational resources utilized by all these techniques are roughly equivalent.

---

### Official Review · Reviewer_up34 · 2023-10-29

**Soundness:** 3 good
**Presentation:** 2 fair
**Contribution:** 2 fair
**Rating:** 5
**Confidence:** 4

**Summary:**

The paper presents REX – a method to achieve more efficient exploration of the action space, leading to a more robust performance in LLM based AI agents.  Through empirical results in blocksworld and GSM8K (math word problems), the approach is shown to achieve superior performances relative to existing approaches.

**Strengths:**

Overall the paper is well motivated and well written. Proposed mechanisms such as integration of rewards into the prompt are novel.

**Weaknesses:**

The approach seems to be limited to problems with discrete actions, and assumes the scores can be mapped to HIGH/LOW which may not always be possible.

**Questions:**

1.	The paper presents two algorithms : REX-UCB and REX-R, which differ only in the way the rewards are classified. I wonder whether these could have been unified into one coherent approach. If not, it is important to explicitly mention when one should use which algorithm.

2.	Is the approach limited to discrete actions? Could it be adapted for continuous action space problems?

3.	Is it assumed that a user can map the UCB scores to HIGH/LOW? This may not always be possible.

4.	Can the method deal with non-binary rewards?

5.	How would the method fare in tasks that involve a long sequence of steps till the final solution is found? Would the LLM’s prediction accuracy (of intermediate steps -with reference to line 7 in Alg 1) be a limitating factor in such tasks?

6.	Can the tasks described be solved with reinforcement learning alone? If so, this would be a natural baseline to add to the experiments.

7.	An intuitive example to accompany 1. And 2. In Section 4.1 would have provided a much clearer picture of the approach.

8.	The introduction could have included more emphasis about how the contributions (1), (2) and (3) are achieved.

9.	In section 4.1 - `form’ should be ‘from’. This mistake is also repeated in later parts of the paper.

---

> ### Author Response · Authors · 2023-11-21
>
> Thank you for your time and valuable feedback! We have tried to address your concerns below:
>
> ---
> Questions:
> ---
>
> **1. The paper presents two algorithms : REX-UCB and REX-R, which differ only in the way the rewards are classified. I wonder whether these could have been unified into one coherent approach. If not, it is important to explicitly mention when one should use which algorithm.**
> * As noted in the paper, REX-R equals REX-UCB when setting the constant $C$ in equation (1) to zero. We introduced these two distinct approaches to highlight the importance of the UCB score over absolute reward. Merely relying on feedback (as Reward) isn't enough for optimal performance; achieving superior results requires a nuanced balance between exploring and exploiting the action space.
> * REX-UCB has showcased better performance compared to REX-R, indicating our preference for REX-UCB across all scenarios. The introduction of REX-R aimed to emphasize that relying solely on Reward for feedback does not lead to optimal performance, as highlighted in the earlier point.
>
> **2. Is the approach limited to discrete actions? Could it be adapted for continuous action space problems?**\
> Yes, in most cases, the approach can be extended to a continuous action space.
> * A straightforward method involves converting the continuous space into a discrete one. However, this approach would yield discrete actions rather than continuous ones.
> * If the continuous action space possesses inherent ordering, such as being all positive real numbers ($\mathbb{R}^+$), the algorithm remains applicable but may introduce some associated error. This reliance largely hinges on the LLM's capacity to discern relative distinctions among various actions. For instance, within the $\mathbb{R}^+$ action space, actions with greater absolute values might yield higher rewards. Consequently, the LLM might learn to predict higher-value actions. Conversely, if cyclic patterns exist between the reward and action space, the LLM might face potential confusion. In summary, REX can definitely be used for continuous action space, but REX's performance in continuous action spaces is contingent upon factors like the environment's nature, the underlying LLM employed, and the complexity of the problem.
>
> **3. Is it assumed that a user can map the UCB scores to HIGH/LOW? This may not always be possible.**\
> No, this assumption was never made. In Algorithm 1, line 5, we can see that the action with the highest UCB score gets 'HIGH' and all the remaining actions get 'LOW'. The user is never involved in any of the REX algorithms.
>
> **4. Can the method deal with non-binary rewards?**\
> Certainly! The current REX implementation supports non-binary rewards. Essentially, our approach utilizes UCB scores, meaning if the UCB formula can handle diverse rewards, our approach can as well.
>
> **5. How would the method fare in tasks that involve a long sequence of steps till the final solution is found? Would the LLM’s prediction accuracy (of intermediate steps -with reference to line 7 in Alg 1) be a limitating factor in such tasks?**\
> Indeed, as outlined in the "Limitations" in Section 5.4 of the paper, our approach relies on In-Context learning, where the input size limitation of the LLM poses a constraint. However, a potential workaround involves segmenting the sequence of actions, summarizing them, and adding them to the prompt. While this solution may partially alleviate the problem, it won't completely resolve it.
>
> **6. Can the tasks described be solved with reinforcement learning alone? If so, this would be a natural baseline to add to the experiments.**\
> Not really. We have taken into account benchmark datasets like GSM8K, where questions are presented in natural language. Thus, relying solely on RL techniques is impractical in this scenario.
>
> **7. An intuitive example to accompany 1. And 2. In Section 4.1 would have provided a much clearer picture of the approach.**\
> Examples are provided in Section C of the Appendix. Happy to share more examples, if you're interested.
>
> **8. The introduction could have included more emphasis about how the contributions (1), (2) and (3) are achieved.**\
> In the Introduction, our emphasis has been on establishing the significance and current trends in Autonomous Agents. We consider it vital to convey the motivation behind our project. The achievement of (1), (2), and (3) is outlined in the final paragraph of the Introduction. We delve deeper into these aspects within the "Proposed Methodology" and "Experiments" sections.
>
> **9. In section 4.1 - `form’ should be ‘from’. This mistake is also repeated in later parts of the paper.**\
> Thank you for helping us identify this typo. We have corrected it.

---

> > ### Comment · Reviewer_up34 · 2023-11-22
> >
> > Thanks for your detailed responses. They have helped me understand your approach better.

---

### Official Review · Reviewer_3Nop · 2023-10-29

**Soundness:** 2 fair
**Presentation:** 2 fair
**Contribution:** 2 fair
**Rating:** 3
**Confidence:** 4

**Summary:**

The authors propose an algorithm called Rapid Exploration and Exploitation (REX) to improve the performance of AI agents while balancing the exploration-exploitation trade-off. REX introduces an additional layer of rewards and integrates concepts similar to Upper Confidence Bound (UCB) scores, leading to more robust and efficient AI agent performance. The authors evaluate the performance of the proposed method in several benchmark tasks.

**Strengths:**

- Important and timely topic
- Nice connections between RL/bandit literature and LLM
- I agree with the authors that MCTS is a promising approach to combine with LLM; hence, the authors try to make progress in a good direction.

**Weaknesses:**

- Regarding experiments, I have several concerns. First, I consider that the authors should compare of the performance of their LLM-based approaches with SOTA methods that are not based on LLMs. Second, the authors state that "RAP is not compatible with OpenAI APIs; therefore, we have exclusively focused on evaluating the time complexity of RAP, without delving into its accuracy", but I do not think that this is a good reason why the authors do not have to empirically evaluate RAP. If the authors emphasize the better computational complexity of their proposed method, they may want to add rigorous theoretical analysis.

- The ideas presented in this paper are quite similar to Tree-of-Thoughts (Yao et al., 2023). Though this is a recent work, but uploaded on arXiv on May 17, 2023; hence, this cannot be regarded as a concurrent work unless I misunderstand the ICLR rule. I would recommend the authors to at least the similarities and differences and compare the authors' proposed method with Yao et al. (2023) in same benchmarks.
    - Yao, Shunyu, et al. "Tree of thoughts: Deliberate problem solving with large language models." arXiv preprint arXiv:2305.10601 (2023).

- Maths are sometimes weird to me. For example:
    - Algorithm 1, line 3: $\forall (s,a) \in Q$. $Q$ is not a set of state-action pairs. This part should be $\forall (s,a) \in S \times A$.
    - Algorithm 1, line 14: $\forall (s,a) \in H$. $H$ is a time step.
- The overall alogrithmic flow is not clear to me. I guess it is mainly because "AGENT.FUNCTION()" (FUNCTION = {SOLVE, VALIDATEACTION, CALCULATEUCB}) is not explained. In addition to the math problems mentioned earlier, the algorithmic flow in this paper is hard to follow.

**Questions:**

[Q1] Could you tell me the similarities and differences between the authors' approach and Tree-of-Thoughts (Yao et al., 2023)? Is it possible to compare the performance of the two methods in the same benchmark? Because the authors use BlocksWorld and GSM8K and Yao et al. (2023) use Game of 24, Creative Writing, and Mini Crosswords, I guess some of them are used for the comparison.

[Q2] Although I can guess what are Agent.Solve and CALCULATEUCB to some extent, there is almost no information on what is Agent.ValidateAction. Could you elaborate more?

[Q3] I suspect that the high performance of REX in BlocksWorld is due to the data leak. Did the authors use the ordinary BlocksWorld environments with default settings?

---

> ### Author Response · Authors · 2023-11-21
>
> Thank you for your time and valuable feedback! We have tried to address your concerns below:
>
> ---
> ## Weaknesses:
> ---
> **First, I consider that the authors should compare of the performance of their LLM-based approaches with SOTA methods that are not based on LLMs.**\
> Our paper focuses on demonstrating the efficiency of MCTS, particularly the UCB scoring mechanism, within LLMs. Comparing with non-LLM methods might not be feasible for datasets where questions are in natural language, making it challenging to adapt them to non-LLM approaches. Dataset nature impacts comparison feasibility; transforming natural language to discrete states for non-LLM methods may not be viable for all datasets. Hence we have only considered LLM-based approaches for comparison.
>
> **Second, the authors state that "RAP is not compatible with OpenAI APIs; therefore, we have exclusively focused on evaluating the time complexity of RAP, without delving into its accuracy", but I do not think that this is a good reason why the authors do not have to empirically evaluate RAP.**\
> We added RAP to demonstrate its slower performance compared to REX in problem-solving. Reflexion and CoT were used as other advanced techniques for performance comparison.
>
> **I would recommend the authors to at least the similarities and differences and compare the authors' proposed method with Yao et al. (2023) in same benchmarks.**\
> Although visually similar, REX and ToT differ in several ways:
> 1. ToT samples possible actions at each state and pursues the most promising action based on voting, while REX directly determines actions from the UCB score, making it more systematic.
> 2. In ToT, an invalid trajectory leads to backtracking and pursuit of the next best trajectory. Conversely, in REX, upon encountering an invalid trajectory, feedback in the form of the UCB score is utilized. In the subsequent iteration, REX pursues the most promising action sequence.
>
> **Algorithm 1, line 3: $\forall (s, a) \in Q$. $Q$ is not a set of state-action pairs. This part should be $\forall (s, a) \in S $x $A$**\
> In Algorithm 1, $Q$ includes all possible combinations of $(s, a)$. Equation (1) shows that to compute the UCB score, every $(s, a)$ pair must have a representation in $Q$. To avoid confusion for the reader, we opted to maintain the expression as "$\forall (s, a) \in Q$".
>
> **Algorithm 1, line 14: $\forall (s, a) \in H$. $H$ is a time step.**\
> As already defined in the "*Require:*" section of Algorithm 1, $H$ represents sequence-of-intermediate-steps. so in line 14, we are conveying the idea that - for every intermediate step taken in the latest solution, we are updating $Q(s, a)$
>
> **The overall alogrithmic flow is not clear to me. I guess it is mainly because "AGENT.FUNCTION()" (FUNCTION = {SOLVE, VALIDATEACTION, CALCULATEUCB}) is not explained. In addition to the math problems mentioned earlier, the algorithmic flow in this paper is hard to follow.**\
> We have added more details about these functions in Section D of the Appendix.
>
> ---
> ## Questions:
> ---
> **[Q1] Could you tell me the similarities and differences between the authors' approach and Tree-of-Thoughts (Yao et al., 2023)? Is it possible to compare the performance of the two methods in the same benchmark? Because the authors use Blocksworld and GSM8K and Yao et al. (2023) use Game of 24, Creative Writing, and Mini Crosswords, I guess some of them are used for the comparison.**
> * We have addressed the similarities and differences between REX and ToT in the above section.
> * (Currently running ToT on Blocksworld and GSM8K, We will share the results soon. )
>
> **[Q2] Although I can guess what are Agent.Solve and CALCULATEUCB to some extent, there is almost no information on what is Agent.ValidateAction. Could you elaborate more?**
> * Details available in Appendix - Section D
>
> **[Q3] I suspect that the high performance of REX in BlocksWorld is due to the data leak. Did the authors use the ordinary BlocksWorld environments with default settings?**\
> * We have confirmed that there's no data leakage in our setup. We created our few-shot examples (3 in total) to form the prompt and used these examples consistently across Reflexion, CoT, and all REX variations. It's impossible for REX to experience data leakage while others do not.

---

> > ### Comment · Reviewer_3Nop · 2023-11-22
> > **Thank you**
> >
> > Thank you for the clarifications. I've read other reviews and rebuttals.
> > I do not have additional questions.

---

> > > ### Author Response · Authors · 2023-11-23
> > >
> > > As a continuation from our previous correspondence, we have completed the experiments using the Tree-of-Thoughts (Yao et al., 2023) technique. The implementation details are as follows: the BFS version of TOT was employed for the experiment, utilizing OAI's GPT-3.5-turbo-0301 LLM. We used value-based action selection in TOT instead of vote-based action selection. Please find the accuracy of various techniques below:
> > >
> > > | Technique | Blocksworld (2-step) | Blocksworld (4-step) | Blocksworld (6-step) | GSM8K |
> > > |-------------|-------------------------|-------------------------|-------------------------|-----------|
> > > | ToT | 50% | 3.2% | 0% | 52.6% |
> > > | REX-R | 53.3% | 37.5% | 14.9% | 81.34% |
> > > | REX-UCB | 80% | 39.28% | 25.43% | 82.03% |
> > > | REX-UCL | 60% | 44.64% | 20.17% | 90.44% |
> > >
> > > The Tree-of-Thoughts (ToT) technique demonstrates strong performance when paired with a language model proficient in Self-Evaluation, such as GPT-4. However, its efficacy diminishes when used with less robust models like gpt-3.5. This discrepancy primarily arises from the higher error rate exhibited by gpt-3.5 when selecting a specific action from the sampled set, which undermines the proposed trajectories within the ToT technique. In contrast, REX, relying on systematic reward collection, exhibits comparatively sound performance even with models like gpt-3.5. The code and the experimental setup have been uploaded for your reference.
> > >
> > > We would be happy to address any additional questions you may have.

---

### Author Response · Authors · 2023-11-23

Dear Reviewers,

We extend our sincerest gratitude to each of you for your invaluable time, effort, and thoughtful feedback provided on our REX paper. Your expertise and constructive insights have significantly enriched our work, contributing immensely to its improvement and refinement. Thank you once again for your diligence, expertise, and dedication to scholarly review.

Throughout this process, we have incorporated supplementary details into our work, aiming to enhance its clarity, available in the Appendix. Furthermore, we conducted a series of experiments utilizing Tree-of-Thoughts (Yao et al., 2023), and the resulting outcomes are presented below. Additionally, we have uploaded the associated code for your reference.

We trust that our discussion has provided you with a clearer understanding of the motivation behind our work, the proposed techniques, and their effectiveness. Should you require further clarification or have additional questions, we are available and eager to engage in further discussion before the Review/Author discussion concludes.

Thank you once again for your valuable contributions to our work!

Best regards,\
Authors of submission 2031

---

### Meta-Review · Area_Chair_4k3G · 2023-12-07

**Metareview:**

The paper introduces REX, an algorithm enhancing Large Language Model (LLM)-based AI agent performance by addressing the exploration-exploitation trade-off. REX, incorporating an additional rewards layer and UCB scores, outperforms existing methods in tasks like blocks world and math world problems. The paper also explores LLM-based reasoning improvements through Monte Carlo Tree Search, presenting two approaches and three evaluated algorithms. These algorithms demonstrate superior accuracy and favorable time complexity compared to certain baseline methods in tasks like block arrangement and grade school mathematics problems.

The paper explores the less-explored but highly relevant topic of the exploration problem in reasoning tasks for Large Language Models (LLMs). Notably, it is easy to read, providing a clear explanation of their method. The motivation behind exploration and exploitation in LLM-based reasoning/agents is effectively communicated, highlighting the paper's significance in addressing a crucial aspect of LLM-based reasoning.

On the other hand, the evaluation falls short in several aspects. It is limited to blocksworld and GSM8K, lacking diverse benchmarks and ablations. Baselines are missing, hindering a thorough comparative analysis. Additionally, implementation details are scarce, impacting reproducibility. Intuition and analytics for REX's operation are needed to clarify its strengths, limitations, and optimal usage scenarios. The paper should provide insights on tuning hyperparameters and offer guidance on when to choose REX over other methods. These enhancements would strengthen the overall evaluation and presentation of REX.

Even if the authors' rebuttals were useful in clarifying some concerns, there is consensus among all reviewers that this paper is not ready for acceptance.

**Justification For Why Not Higher Score:**

The consensus among the reviewers indicates that the paper is not ready for acceptance.

**Justification For Why Not Lower Score:**

N/A

---

### Decision · Program_Chairs · 2024-01-16

Reject